# Cognitive Status Epilepticus: Two Case Reports

**DOI:** 10.3390/medicina57080799

**Published:** 2021-08-03

**Authors:** Eleni Karantali, Symela Chatzikonstantinou, Ioannis Mavroudis, Constantin Trus, Dimitrios Kazis

**Affiliations:** 1Third Neurological Department, Faculty of Medicine, Aristotle University of Thessaloniki, 57010 Thessaloniki, Greece; melina.chatzik@gmail.com (S.C.); dimitrios.kazis@gmail.com (D.K.); 2Department of Neurosciences, Leeds Teaching Hospitals NHS Trust, Leeds LS97TF, UK; i.mavroudis@nhs.net; 3Department of Morphological and Functional Sciences, Faculty of Medicine, Dunarea de Jos University, 800008 Galati, Romania

**Keywords:** status epilepticus, amnesia, aphasia, EEG

## Abstract

Cognitive status epilepticus is an uncommon form of focal status epilepticus presenting with a dysfunction of language, thinking or associated higher cortical functions. The absence of ictal manifestations can be misleading and delay a prompt diagnosis. Here we present two patients; one with amnesic and one with aphasic status epilepticus. Through these cases, we aim to highlight the value of EEG performance early in the diagnostic work-up and early antiepileptic drug initiation in cases where an epileptic disorder cannot be excluded.

## 1. Introduction

Focal status epilepticus (FSE) with or without an impairment of consciousness can be presented with motor or non-motor symptoms; the latter was formerly known as aura continua [1,2]. Typically, the duration is over an hour but on several occasions it can last for years. EEG findings represent locally restricted epileptic activity. FSE, when presented with non-motor symptoms, may be very challenging to diagnose.

In the present study, we describe the cases of two patients with cognitive status epilepticus; the first suffering from amnestic and the second from aphasic status epilepticus. These cases address the difficulty of diagnosing such rare clinical entities.

## 2. Case Presentation

### 2.1. Patient One; Amnestic SE

A 49-year-old right-handed female patient presented with an amnestic disorder that started upon waking. Three months prior, a similar incident was reported, which lasted for a day and was spontaneously resolved. A neurological examination revealed anterograde and autobiographical amnesia. The patient was afebrile and otherwise cooperative and relatively calm. The emergency brain non-contrast computed tomography (NCCT) was normal. Considering the patient’s personal history, we conducted an emergent scalp EEG, which revealed continuous spike-and-wave complexes 3–3.5 Hz/s over the left hemisphere, which consequently confirmed the diagnosis of amnestic status epilepticus (Figure 1A). No other ictal features were seen. The patient was treated with intravenous benzodiazepines and valproic acid. Two days later, we performed a second EEG because of a residual amnestic disorder, which revealed very frequent spike-and-wave activity (Figure 1B). The patient commenced brivaracetam as an add-on therapy and her symptoms and EEG findings resolved completely. When completely improved, the patient recollected all the incidence of her amnestic phase (Figure 1C). A brain MRI scan was performed within 24 h from the symptom onset and it was reported as being unremarkable. Two months later, on follow-up, the patient remained seizure-free.

### 2.2. Patient Two; Aphasic SE

A 48-year-old right-handed female patient presented with acute onset speech difficulties upon waking. During the previous two weeks, similar episodes with a shorter duration (few minutes) and a spontaneous remission were reported. Her past medical history included a chronic ischemic stroke with mild residual aphasia and right hemiparesis, essential hypertension, hyperhomocysteinemia and hypercholesterolemia. A neurological examination revealed receptive aphasia, right spastic hemiparesis and mild psychomotor agitation. The patient was afebrile. A brain NCCT revealed a hypoattenuation of left temporal lobe in keeping with a chronic stroke. The following day, a scalp EEG was performed to exclude symptomatic epilepsy. No other ictal features were noticed. Continuous periodic epileptiform discharges (PEDs) on the left temporal region were noted, confirming the diagnosis of focal aphasic status epilepticus (Figure 2A). After the initiation of levetiracetam followed by the addition of valproic acid, the patient gradually improved (Figure 2B). A brain MRI was performed within 24 h from symptom onset and revealed a chronic ischemic stroke of the left temporal lobe. The rest of the diagnostic work-up was unremarkable. On follow-up, two months later, the patient remained seizure-free.

## 3. Discussion

Based on ILAE epilepsy classification, the term focal status epilepticus (FSE) with or without an impairment of consciousness is used to describe the entity formerly known as partial status epilepticus [1]. The typical presentation is characterized by prominent motor or sensorimotor symptoms. Rarely, focal status epilepticus can present with a wide variety of non-motor symptoms with autonomic, cognitive, sensory, emotional or behavioral arrest [1]. The minimum duration is reported to be over an hour with subtle motor symptoms that can occur at time intervals not longer than 10 s whereas the symptoms of non-motor FSE may be waxing and waning, continue uninterrupted and persist during sleep [2]. In most patients with motor FSE, the impairment of consciousness can be safely assessed. In contrast, a reliable assessment of consciousness in non-motor FSE is particularly challenging especially when patients exhibit language or memory dysfunction [3].

Cognitive FSE presents with a dysfunction of language, thinking or associated higher cortical functions [1]. These symptoms outbalance the presence of other epileptic manifestations. In non-convulsive status epilepticus (NCSE), as the clinical signs and symptoms are usually non-specific or diagnostic, the diagnosis is based on the EEG performance and the initiation of an antiseizure drug. EEG patterns can be divided into diagnostic (e.g., focal or generalized spikes, sharp waves or sharp-and-slow complexes) or uncertain significance (e.g., periodic or rhythmic discharges) [4]. When the clinical suspicion of NCSE is high but the EEG findings are non-diagnostic, a trial of an IV antiseizure drug is suggested [4].

Patient One presented with acute onset memory deficits and underlying focal spike-wave complexes. Amnestic status epilepticus is a rare form of cognitive FSE. The differentiation from transient global amnesia (TGA) can be particularly challenging. TGA presents with an acute onset global (retrograde and anterograde) memory deficit lasting less than 24 h. Other cognitive functions are typically normal. A single, definite etiology has yet to been determined. The underlying pathophysiologic processes that have been proposed include vascular, migraine, epileptic and psychogenic mechanisms. Transient epileptic amnesia (TEA) can be presented as TGA. Typically, TEA episodes are shorter, recurrent and, occasionally, accompanied by ictal manifestations such as oral automatisms or hallucinations [5]. From a clinical point of view, the recurrence and atypical symptoms such as confusion are the key to discriminating between TEA and TGA [5]. The diagnostic value of an EEG in TGA may be limited. After a TGA episode, the EEG is typically normal. On the contrary, an EEG in TEA is usually diagnostic. In patients presenting with atypical TGA, an EEG must be performed. In our patient, the recurrent amnestic syndrome led us to perform an EEG and diagnose the amnestic status epilepticus. In general, EEG findings, a previous history of similar episodes, an atypical presentation and the improvement of symptoms after the initiation of AEDs may contribute to the diagnosis.

Patient Two presented with an acute onset aphasia, which typically can be seen in cerebrovascular events. Rarely, an acute onset aphasia can be the manifestation of epileptic activity. In the absence of typical paroxysmal seizure symptoms, the differential diagnosis solely on a clinical basis can be improbable. Both ictal aphasia and aphasic status epilepticus vary in phenomenology (paraphasia to mutism) and EEG findings [3]. The initiation of AEDs may be indicated in cases where epileptic aphasia cannot be excluded.

In conclusion, our cases highlight the difficulties of diagnosing cognitive status epilepticus on a clinical basis and stress the value of performing an EEG early in the diagnostic work-up.

## Figures and Tables

**Figure 1 medicina-57-00799-f001:**
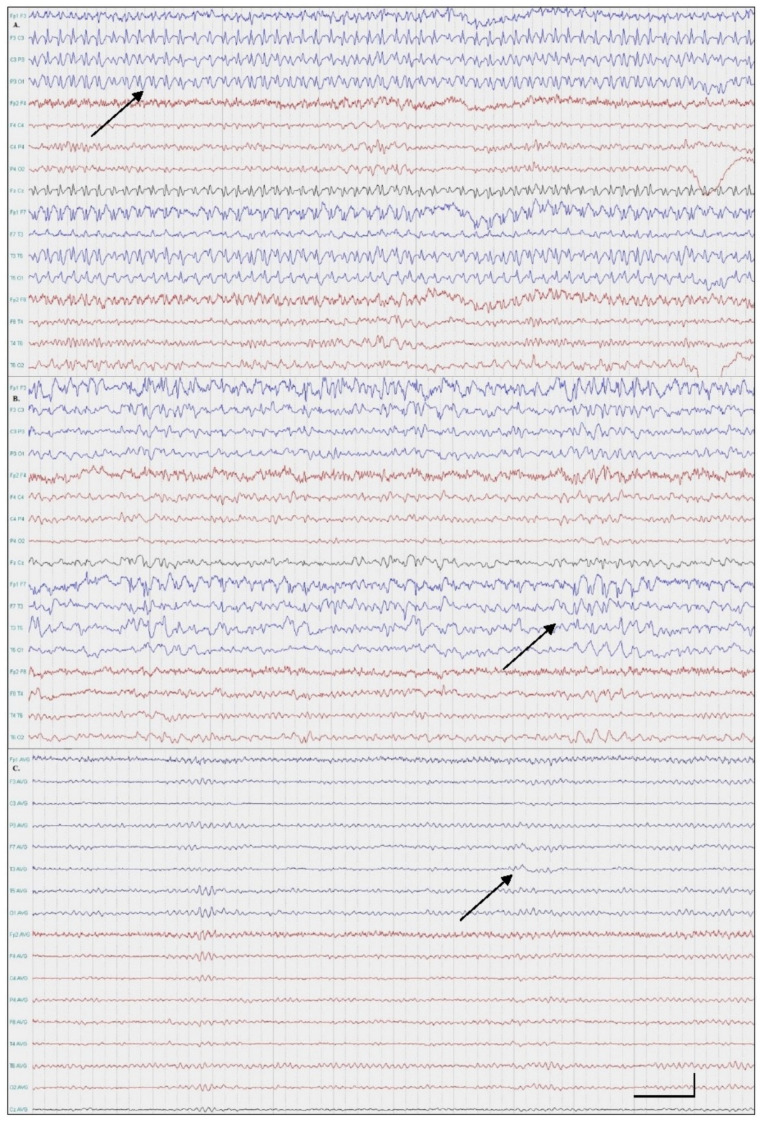
EEGs: Patient One. (**A**) Continuous spike-wave activity over the left hemisphere (maximum over frontotemporal areas); (**B**) very frequent spike-wave activity (double banana montage, 0.3–70 Hz); (**C**) Intermittent theta wave activity over the left frontotemporal region (average monopolar montage, 0.3–70 Hz).

**Figure 2 medicina-57-00799-f002:**
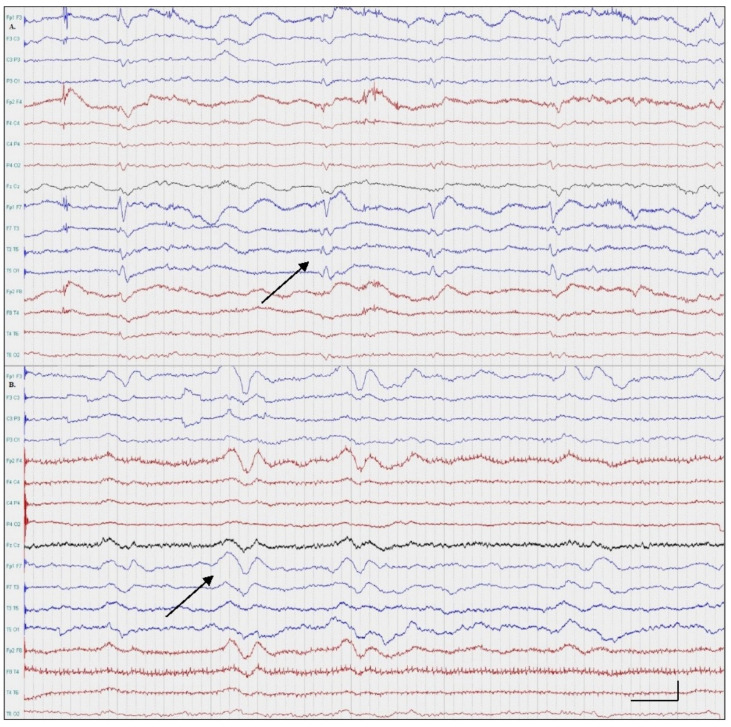
EEGs: Patient Two. (**A**) Continuous periodic epileptiform discharges over the left temporal region; (**B**) very frequent polymorphic theta and delta slow wave activity over the left hemisphere (double banana montage, 0.3–70 Hz).

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
