# Peer review of "Cognitive Status Epilepticus: Two Case Reports"

_medicina, 2021, doi:10.3390/medicina57080799_

Round 1

Reviewer 1 Report

  1. It is not so precise in some medical terms. For example: automatic remission -> spontaneous remission or self-limiting; spike-wave -> spike-and-wave.
  2. Some important clinical personal information is missing, including gender and handedness. In the second case, we don’t have any clues of dominant hemisphere. Therefore, we don’t have strong correlation between aphasia and periodic discharges over left cerebral hemisphere.
  3. When did the patients have his/her MRIs? What is the duration between MRIs and EEG/clinical onset? In case 2, periodic discharges < 2.5 Hz not always suggested status epilepticus. Other differential diagnosis included acute brain injury, e.g. acute stroke. Although the authors mentioned the aphasia was improving gradually after usage of ASM, aphasia from stroke also getting better gradually. Therefore, the timing of brain MRI was important. No acute stroke in MRI in acute stage could exclude this differential diagnosis.
  4. The arrows are pointing incorrectly in Figure 1 B and C.

Author Response

It is not so precise in some medical terms. For example: automatic remission -> spontaneous remission or self-limiting; spike-wave -> spike-and-wave.

Response: Thank you for your comments. We have corrected the medical terms you pointed out.

Some important clinical personal information is missing, including gender and handedness. In the second case, we don’t have any clues of dominant hemisphere. Therefore, we don’t have strong correlation between aphasia and periodic discharges over left cerebral hemisphere.

Response: Thank you for your feedback. We have included some additional information for both patients. The second patient was right-handed, thus the aphasia is correlated with the EEG findings.

When did the patients have his/her MRIs? What is the duration between MRIs and EEG/clinical onset? In case 2, periodic discharges < 2.5 Hz not always suggested status epilepticus. Other differential diagnosis included acute brain injury, e.g. acute stroke. Although the authors mentioned the aphasia was improving gradually after usage of ASM, aphasia from stroke also getting better gradually. Therefore, the timing of brain MRI was important. No acute stroke in MRI in acute stage could exclude this differential diagnosis.

Response: Thank you for your constructive feedback. You are totally right. The brain MRI was conducted within 24hours after the symptom onset. Only the chronic stroke on the left temporal lobe was depicted. We also included a specific phrase in our text: "Brain MRI was performed within 24 hours from the symptom onset and revealed a chronic ischemic stroke of the left temporal lobe."

The arrows are pointing incorrectly in Figure 1 B and C.

Response: Thank you for your comment. We have corrected both figures.

Reviewer 2 Report

A clear and succinct article describing cogntivie status epilepticus in two patients.

Author Response

Response: Thank you for your kind words.

Round 2

Reviewer 1 Report

  1. The authors answered my questions well. However, there are still lots of problems in English wording. I strongly suggest this manuscript need new English editing.
  2. In the discussion, there is a sentence of “In the absence of ictal manifestations, differential diagnosis from transient global amnesia (TGA) can be particularly challenging.”. Did you mean that it is difficult to differentiate amnestic SE from TGA? However, TGA is a syndrome and one of the possible causes of TGA is seizure. Therefore, this sentence is confusing. Please clarify it.
  3. The authors stressed out the value of EEG in the conclusion. However, it was not enough to make diagnosis of SE by only the EEG in case 2. It need to combine clinical information (including the improvement after AEDs use) with EEG. It is better to add the discussion about the diagnostic criteria of EEG of NCSE to make the discussion more in depth.
  4. Lack of “Institutional Review Board Statement” and “Informed Consent Statement”.

Author Response

The authors answered my questions well. However, there are still lots of problems in English wording. I strongly suggest this manuscript need new English editing.

Response: Thank you for your feedback. Our manuscript was proofread by an English-speaking editor.

In the discussion, there is a sentence of “In the absence of ictal manifestations, differential diagnosis from transient global amnesia (TGA) can be particularly challenging.”. Did you mean that it is difficult to differentiate amnestic SE from TGA? However, TGA is a syndrome and one of the possible causes of TGA is seizure. Therefore, this sentence is confusing. Please clarify it.

Response: Thank you for your comment. You are totally right. We further analyzed the above sentence in our Discussion section.

The authors stressed out the value of EEG in the conclusion. However, it was not enough to make diagnosis of SE by only the EEG in case 2. It need to combine clinical information (including the improvement after AEDs use) with EEG. It is better to add the discussion about the diagnostic criteria of EEG of NCSE to make the discussion more in depth.

Response: Thank you for your feedback. We added a specific section discussing about EEG diagnostic criteria of NCSE.

Lack of “Institutional Review Board Statement” and “Informed Consent Statement”.

Response: Thank you for your comment. We added both statements.